# Hypercalcemia in Pregnancy Due to CYP24A1 Mutations: Case Report and Review of the Literature

**DOI:** 10.3390/nu14122518

**Published:** 2022-06-17

**Authors:** Stefan Pilz, Verena Theiler-Schwetz, Pawel Pludowski, Sieglinde Zelzer, Andreas Meinitzer, Spyridon N. Karras, Waldemar Misiorowski, Armin Zittermann, Winfried März, Christian Trummer

**Affiliations:** 1Department of Internal Medicine, Division of Endocrinology and Diabetology, Medical University of Graz, Auenbruggerplatz 15, 8036 Graz, Austria; verena.schwetz@medunigraz.at (V.T.-S.); christian.trummer@medunigraz.at (C.T.); 2Department of Biochemistry, Radioimmunology and Experimental Medicine, The Children’s Memorial Health Institute, 04-730 Warsaw, Poland; p.pludowski@ipczd.pl; 3Clinical Institute of Medical and Chemical Laboratory Diagnostics, Medical University of Graz, 8036 Graz, Austria; sieglinde.zelzer@medunigraz.at (S.Z.); andreas.meinitzer@medunigraz.at (A.M.); winfried.maerz@synlab.de (W.M.); 4National Scholarship Foundation, 55535 Thessaloniki, Greece; karraspiros@yahoo.gr; 5Department of Endocrinology, Centre of Postgraduate Medical Education, Bielanski Hospital, 01-809 Warsaw, Poland; wmisiorowski@cmkp.edu.pl; 6Clinic for Thoracic and Cardiovascular Surgery, Herz- und Diabeteszentrum Nordrhein-Westfalen (NRW), Ruhr University Bochum, 32545 Bad Oeynhausen, Germany; azittermann@hdz-nrw.de; 7SYNLAB Academy, Synlab Holding Deutschland GmbH, 68159 Mannheim, Germany; 8V th Department of Medicine (Nephrology, Hypertensiology, Rheumatology, Endocrinology, Diabetology, Lipidology), Medical Faculty Mannheim, University of Heidelberg, 68167 Mannheim, Germany

**Keywords:** vitamin D, CYP24A1, pregnancy, hypercalcemia, fertility, lactation, intoxication, supplementation, idiopathic infantile hypercalcemia

## Abstract

Pathogenic mutations of CYP24A1 lead to an impaired catabolism of vitamin D metabolites and should be considered in the differential diagnosis of hypercalcemia with low parathyroid hormone concentrations. Diagnosis is based on a reduced 24,25-dihydroxyvitamin D to 25-hydroxyvitamin D ratio and confirmed by genetic analyses. Pregnancy is associated with an upregulation of the active vitamin D hormone calcitriol and may thus particularly trigger hypercalcemia in affected patients. We present a case report and a narrative review of pregnant women with CYP24A1 mutations (13 women with 29 pregnancies) outlining the laboratory and clinical characteristics during pregnancy and postpartum and the applied treatment approaches. In general, pregnancy triggered hypercalcemia in the affected women and obstetric complications were frequently reported. Conclusions on drugs to treat hypercalcemia during pregnancy are extremely limited and do not show clear evidence of efficacy. Strictly avoiding vitamin D supplementation seems to be effective in preventing or reducing the degree of hypercalcemia. Our case of a 24-year-old woman who presented with hypercalcemia in the 24th gestational week delivered a healthy baby and hypercalcemia resolved while breastfeeding. Pathogenic mutations of CYP24A1 mutations are rare but should be considered in the context of vitamin D supplementation during pregnancy.

## 1. Introduction

Accurate diagnosis and treatment of hypercalcemia during pregnancy is crucial because it is associated with an increased risk of adverse health outcomes for the mother and the fetus [1,2,3]. Hypercalcemia during pregnancy is relatively rare and its further differential diagnosis is dependent on the prevailing parathyroid hormone (PTH) level. Primary hyperparathyroidism (PHPT) that is characterized by elevated or inappropriately high parathyroid hormone (PTH) concentrations seems to be the main cause of hypercalcemia during pregnancy [1,2,3]. Pregnant women with hypercalcemia and reduced PTH concentrations may suffer from a variety of diseases including, e.g., disorders of vitamin D metabolism, vitamin D intoxication per se, malignancy, granulomatous diseases (sarcoidosis and tuberculosis), pseudohyperparathyroidism (due to elevated PTH related peptide (PTHrP) levels), milk-alkali syndrome (due to excess intake of calcium and antacid drugs), etc. posing a diagnostic and therapeutic challenge [2,3,4,5,6].

In 2011, Schlingmann et al. used a candidate gene approach to evaluate the cause of idiopathic infantile hypercalcemia, a condition that had a particular high incidence in Great Britain in the 1950s during a time of high vitamin D supplementation and food fortification [7]. They identified loss-of-function mutations in cytochrome-P450 family 24 subfamily A member 1 (CYP24A1) with an indication for autosomal recessive inheritance in affected children who were particularly prone to develop hypercalcemia after high dose vitamin D supplementation [7]. This CYP24A1 gene encodes the mitochondrial inner membrane P450 24-hydroxylase enzyme responsible for the catabolism of vitamin D metabolites [7,8,9]. It catalyzes the conversion of 25-hydroxyvitamin D (25(OH)D), the main circulating vitamin D metabolite that is used for the classification of vitamin D status, and of 1α,25-dihydroxyvitamin D (1,25(OH)_2_D), also called calcitriol or the active vitamin D hormone, to inactive (or less active) metabolites [8,9]. In detail, 25(OH)D is converted to 24,25-dihydroxyvitamin D (24,25(OH)_2_D) and 1,25(OH)_2_D is converted to 1,24,25-trihydroxyvitamin D as an initial step of their catabolism [8,9].

Most patients with idiopathic infantile hypercalcemia probably remain unrecognized as there are only a few hundred cases described in the literature (i.e., 221 cases in a recent systematic review) with a Polish study suggesting a population disease frequency of 1 in 32,465 births [4,10]. Of note, the term idiopathic infantile hypercalcemia is meanwhile considered a misnomer for patients with pathogenic mutations of CYP24A1 as these patients can develop hypercalcemia across their whole life-span [4]. Due to the increasing use of vitamin D supplements and food fortification, knowledge on pathogenic CYP24A1 mutations in terms of their clinical relevance are of public health interest. Importantly, pregnancy may be a trigger factor for the development of hypercalcemia in affected patients with pathogenic CYP24A1 mutations, but data on this topic are extremely rare and there exists a knowledge gap regarding guidance for the management of affected patients in this setting [4].

In this work, we present the case of a pregnant woman with hypercalcemia due to pathogenic mutations in CYP24A1. In addition, we perform a literature review to summarize the current knowledge on this issue by outlining the laboratory and clinical characteristics during pregnancy and postpartum and the applied treatment approaches. Based on the totality of available evidence we aim to provide some guidance for the management of affected women during pregnancy, postpartum and lactation.

## 2. Case Report

In summer 2021, a 24-year-old pregnant woman at 24 weeks of gestation was referred to our outpatient clinic of the Department of Endocrinology & Diabetology of the Medical University of Graz, Austria, for further diagnostics and treatment of hypercalcemia. A total serum calcium of 3.5 mmol/L had been incidentally discovered at a routine laboratory measurement about 1 week before the initial visit in our outpatient clinic. At that time, she had complained of polyuria, increased thirst and fatigue, and hydronephrosis grade 2 to 3 on the left side had been diagnosed at an external hospital, followed by an insertion of a double J catheter. Kidney stones or nephrolithiasis had, however, not been detected at any time in her life. She was already advised to stop the intake of a multimicronutrient supplement containing 800 international units (IU) of vitamin D3 (20 µg) that she had taken daily until then.

At our outpatient clinic, she reported no symptoms and no major previous diseases, except, an abortion in the 6th week of gestation two years before. She took a daily iron and magnesium supplement, and levothyroxine (LT4) 75µg daily had been prescribed about 1 week before due to hypothyroxinemia with subsequently normal thyroid function tests throughout pregnancy with this treatment. Her laboratory report yielded hypercalcemia with reduced PTH concentrations, and we recommended the avoidance of vitamin D supplements and sun exposure, low calcium intake and high oral fluid intake, and arranged another appointment for a more extensive laboratory work-up (see Table 1 for selected laboratory results at our department with laboratory methods as described in previous publications [11,12,13,14]).

Even after the follow-up appointment, we could not identify a causal disease for her PTH-independent hypercalcemia. We did not have an indication for tuberculosis (normal gamma-interferon test), for sarcoidosis (normal soluble interleukin-2 receptor and angiotensin-converting-enzyme (ACE)), for pseudohyperparathyroidism or malignancy (normal PTHrP), for familial hypocalciuric hypercalcemia (FHH) (hypercalciuria and low PTH) and no clear clinical and laboratory indication for hereditary hypophosphatemic rickets with hypercalciuria (HHRH) (tubular resorption of phosphate: 80% to 90%), or other diseases (a magnetic resonance imaging of the chest and abdomen was recommended by us but not performed). We advised to start with a potassium supplement (i.e., Reducto-spezial^®^ 3 times 1 tablet daily) due to its potential to reduce intestinal calcium absorption, increased her magnesium supplement dose and made another (3rd) appointment to test for vitamin D metabolites in order to evaluate for the presence of 24-hydroxylase deficiency caused by pathogenic mutations of CYP24A1. As she had a reduced 24,25-dihydroxyvitamin D to 25(OH)D ratio, we made a final (4th) appointment at our outpatient clinic for genetic analyses and confirmed that she was compound heterozygous for two previously described pathogenic variants of CYP24A1, i.e., c.443T > C (p.Leu148Pro) and c.1226T > C (p.Leu409Ser). At 40 weeks of gestation, labor was induced in an external hospital due to suspected fetal macrosomia and the patient delivered a healthy daughter (4160 g and 54 cm) with a normal total serum calcium of 2.33 mmol/L and 2.25 mmol/L on day 2 and 4 of her life, respectively, and an uncomplicated normal development thereafter. Maternal total serum calcium concentrations were 2.67, 2.60, and 2.77 mmol/L on the day of giving birth, one and three days thereafter, respectively. Breastfeeding was started and continued by the mother until the last appointment at our department, more than two months after giving birth. At that appointment, she presented symptom free with normal serum calcium concentrations. The decision to not refrain from breastfeeding as most previous cases was partially based on the clinical experience of one of our co-authors from Poland (W.M.) that women with pathogenic CYP24A1 mutations showed a gradual decline in serum calcium concentrations while breastfeeding (unpublished observation). Usual vitamin D supplementation with 400 IU per day was recommended for the newborn daughter and she has been developing well without any complications. Follow-up laboratory tests in a few months were recommended for the mother and the newborn to check for serum calcium and creatinine levels.

The patient gave written informed consent for this publication and was included in the Graz Endocrinology Registry Study that was approved by the ethics committee at the Medical University of Graz, Austria.

## 3. Literature Review

A literature review in PubMed using the search terms “CYP24A1” and “pregnancy” was performed on 13 April 2022 to identify articles presenting data on pregnant women with pathogenic CYP24A1 mutations causing hypercalcemia. Out of 97 articles, we identified 9 eligible manuscripts by this search strategy and by screening their reference lists, we retrieved two additional publications so that 11 articles were included in our work [15,16,17,18,19,20,21,22,23,24,25]. Overall, we retrieved 13 women with pathogenic CYP24A1 mutations who had reports on 29 pregnancies. Selected characteristics of the pregnancies of the affected women are presented in Table 2. Obstetric complications were frequently reported but the vast majority of women delivered a healthy baby with usually either no or just transient complications, mainly due to disturbed calcium metabolism. We could not identify a clear pattern regarding changes of the magnitude of hypercalcemia as a function of pregnancy time. Due to the low number of cases and pregnancies we refrained from performing any statistical analyses to estimate relative risks for obstetric complications in women with pathogenic CYP24A1 mutations as opposed to the general pregnancy population.

Regarding treatment of hypercalcemia during pregnancy and postpartum there are no randomized controlled trials (RCTs) available so that any treatment approach was only based on a risk benefit estimation and mainly on expert opinions. The general treatment recommendation for most patients was, of course, to strictly avoid vitamin D supplementation and also to minimize sunlight (ultraviolet-B) induced vitamin D synthesis in the skin. Avoidance of vitamin D supplements during pregnancy in affected women seems to have a great effect on serum calcium. In this context, one woman used vitamin D supplements during her first pregnancy and had serum calcium concentrations up to 3.3 mmol/L, but remained normocalcemic throughout her second pregnancy during which she avoided vitamin D supplements [24]. Nevertheless, several pregnant women developed hypercalcemia during pregnancy despite avoidance of vitamin D supplement intake [17]. In addition, a high oral fluid intake and dietary calcium restriction were usually advised during pregnancy. Given that total serum calcium levels decrease during pregnancy as a consequence of reduced serum albumin levels due to volume expansion, it is generally recommended to measure albumin adjusted and/or ionized (free) calcium to assess calcium levels in pregnancy. Specific drug treatments of these patients during pregnancy are shown in Table 3.

We assumed that there was no specific treatment when it was not clearly indicated in the respective publication. During 29 pregnancies identified in the literature, the applied treatment approaches were intravenous hydration in four, loop diuretics in three, glucocorticoids in four, phosphate supplements in two (in one case in combination with omeprazole) and calcitonin in two pregnancies, respectively. Conclusions on efficacies of these treatments cannot be made due to the observational nature of these limited data and the challenge to disentangle potential effects of specific drugs (a) from each other when used in combination, (b) from general treatment approaches such as stopping vitamin D supplement intake, and (c) from the natural course of the disease. In general, serum calcium levels were not significantly and consistently reduced or normalized after initiation of the above-described specific drug treatments except for some cases in which vitamin D supplement intake was stopped in parallel (e.g., case numbers 6 and 9 in Table 3) [19,21]. Of note, some authors concluded on the missing effect of their treatment such as in terms of glucocorticoids (case number 13 in Table 3) [24]. Glucocorticoids are known to modulate vitamin D metabolism resulting in lower calcitriol levels by effects including induction of 24-hydroxylation and reduce intestinal calcium absorption, but there are still several knowledge gaps regarding this issue. Importantly, glucocorticoids may not be effective in treating hypercalcemia in patients with CYP24A1 mutations as reported previously for non-pregnant patients [26]. It should also be noted that there was no use of bisphosphonates or denosumab during pregnancy, probably because bisphosphonates cross the placenta and because there is hardly any experience with denosumab during human pregnancy. Data from case reports on the use of bisphosphonates before conception or during pregnancy do not report major teratogenic effects, but some cases of transient neonatal hypocalcemia and low birth weight occurred [27,28,29,30]. Calcitonin does not cross the placenta but its calcium lowering effects diminish after a few days due to tachyphylaxis (i.e., downregulation of its receptors on osteoclasts). Specific drug treatments after delivery (i.e., postpartum) are outlined in Table 4.

Regarding the specific treatments of the women after delivery there were 11 cases treated with intravenous hydration, three cases treated with loop diuretics, four cases treated with glucocorticoids, five cases treated with calcitonin, two cases treated with denosumab and four cases treated with bisphosphonates. Conclusions on the efficacy of these treatments are limited based on the observational data with its inherent limitations as noted above, and in particular due to the natural course of serum calcium fluctuations postpartum. In this context, some but not all, reports documented an increase in serum calcium within the first days to few weeks after delivery. Following this immediate postpartum period, a gradual improvement, i.e., reduction of serum calcium concentrations, was reported over the next several weeks to a few months in virtually all cases with available data. Interestingly, in two cases, denosumab treatment in the postpartum period was followed by a significant decrease in serum calcium concentrations after only a few days, while preceding treatments with, e.g., bisphosphonates, furosemide and glucocorticoids were insufficient to control hypercalcemia (case numbers 4 and 5 in Table 4) [18].

The vast majority of newborns of mothers with hypercalcemia due to CYP24A1 mutations in pregnancy is not affected by idiopathic infantile hypercalcemia, as this disease typically follows an autosomal recessive inheritance pattern. In the newborns, hypercalcemia was detected in some cases that usually resolved after several days to a few weeks as well as transient hypocalcemia, and some cases reported on transient hypoglycemia (see Table 4). One newborn was compound heterozygous for pathogenic CYP24A1 mutations and developed symptomatic hypercalcemia after receiving 50,000 IU of vitamin D2 on day 1 of his life [15]. It should also be noted that the clinical significance of heterozygote carriers is not clear at present [8]. Some publications report on a mild, yet clinically significant, phenotype of some heterozygote carriers with slight hypercalcemia and hypercalciuria [8]. Therefore, common vitamin D doses for rickets prevention, e.g., 400 IU per day, should not be exceeded in these newborns and screening for hypercalcemia may be prudent as recommended for our case.

## 4. Discussion

We have presented a case report and results of a systematic literature review on pregnant women with pathogenic CYP24A1 mutations. It is evident that pregnancy with its associated changes in vitamin D metabolism triggers or enhances hypercalcemia in affected women. Obstetric complications (e.g., arterial hypertension, pre-eclampsia and hypercalcemic crisis) were frequently reported, but most pregnancies resulted in live births (see Table 2). Although there were also some, usually minor, complications in the newborns (e.g., hypercalcemia, hypocalcemia or hypoglycemia) immediately after birth (see Table 2), we can conclude that the vast majority of newborns will be healthy and have a normal development in their further life as reported by most of the cases.

In general, pathogenic CYP24A1 mutations are relatively rare but in view of the high prevalence of vitamin D supplementation during preconception and pregnancy, affected women are at particularly high risk of severe hypercalcemia and associated obstetrics complications. From a pathophysiologic point of view, pregnancy is associated with about a doubling to tripling of serum 1,25(OH)_2_D (calcitriol) concentrations that seems to be important to increase intestinal calcium absorption in order to meet the mineral (calcium) demands of the growing fetus [1,31]. Therefore, it appears logical that in this setting of increased “vitamin D activation”, i.e., hydroxylation of 25(OH)D to 1,25(OH)_2_D, a disease with impaired vitamin D catabolism, may be aggravated in pregnancy. In this context, we support the suggestion of a European expert consensus to measure serum calcium concentrations as part of otherwise indicated routine screening programs or visits during preconception and early pregnancy [1]. Such an approach targets to detect alterations in calcium metabolism that may, beyond CYP24A1 mutations, of course, be important to diagnose parathyroid disorders and related diseases. In the case of pathogenic CYP24A1 mutations, it appears logical to assume that the degree of hypercalcemia may be, as in the case of PHPT, associated with the risk of obstetric complications although we cannot definitely claim this due to relatively few reported cases. In this context, it is well known from patients with PHPT and other related diseases that hypercalcemia per se can cause glomerular hyperfiltration with hypercalciuria and increased risk of kidney stones (urolithiasis/nephrolithiasis), worsening of kidney function or pancreatitis. These latter complications have also been reported by some of the cases (see Table 2). During pregnancy, calcium is transported by the placenta to the fetus with the consequence that hypercalcemia of the mother is also causing hypercalcemia of the fetus. This may in turn suppress PTH in the fetus and may explain why newborns of mothers with CYP24A1 mutations are at risk of both, hypercalcemia due to transfer of calcium from the mother during pregnancy and of hypocalcemia due to suppression of PTH. Therefore, we suggest, as it is recommended for newborns of mothers with PHPT, to measure serum or ionized calcium concentrations in the newborns every second day starting on day two for about 1 to 2 weeks [1]. Considering the uncertainty regarding the clinical significance of heterozygote carriers for CYP24A1 mutations it may also be justified to re-check the serum calcium concentrations in the children again several weeks to a few months after birth, in order to capture hypercalcemic episodes that may be triggered by usual vitamin D supplementation. It may also appear reasonable to measure blood glucose levels in the newborns in the first days after birth as some cases of hypoglycemia have been reported [17]. Serum calcium concentrations should also be measured in the affected mothers in the first week after giving birth (with re-measurements of serum calcium depending on the initial value) as the calcium transfer via the placenta is immediately stopped after delivery and may thus further aggravate hypercalcemia in the mother.

Regarding treatment approaches during pregnancy, it is logical and also seems to be highly effective that women with pathogenic CYP24A1 mutations are advised to stop any vitamin D supplementation and aim to minimize other sources of vitamin D supply. In particular sunlight (ultraviolet-B (UV-B)) induced vitamin D synthesis in the skin should be avoided. Notably, despite our advice to minimize vitamin D supply in our case patient there were no major changes in serum calcium concentrations, but we noticed a significant decrease in urinary calcium excretion. Data interpretation of this single case is, however, challenging as we, of course, have no data on the natural course of the disease without any intervention. In addition, a sufficient oral fluid intake and avoidance of overwhelming calcium supply can be recommended. Intravenous hydration has been used as a treatment approach for several women with pathogenic CYP24A1 mutations during pregnancy and postpartum, but although this treatment is generally safe and efficient for treatment of hypercalcemia, risk of volume overload (edema) should be kept in mind [32]. Apart from this, other specific drug treatments should only be used on an individual basis during pregnancy by considering and balancing the potential risks and benefits. Overall, the existing literature does not clearly support the efficacy of any of these treatments so that we would rather be cautious when using them during pregnancy. In our case woman, we would have initiated a specific drug treatment for hypercalcemia in case of albumin adjusted serum calcium concentrations above 3.5 mmol/L. We personally consider albumin adjusted serum calcium concentrations from 3.0 to 3.5 mmol/L as a range in which specific drug treatments to lower calcium levels should be seriously considered if other measures are not effective and if symptoms or adverse consequences of hypercalcemia emerge. Breastfeeding was not established in most women after delivery as lactation might potentially aggravate hypercalcemia. In our case report we did, however, observe a gradual decline in serum calcium concentration while breastfeeding. Therefore, it appears reasonable that breastfeeding should not be a priori banned in women with pathogenic CYP24A1 mutations. Regarding treatment options for severe hypercalcemia in the postpartum period, denosumab appeared to be effective in reducing serum calcium concentrations.

Beyond pregnancy and the postpartum period, data on the long-term perspective of patients with pathogenic CYP24A1 mutations suggest that affected patients may be at increased risk of chronic kidney disease, nephrocalcinosis and nephrolithiasis (kidney stones) despite avoidance of sun exposure, vitamin D and calcium supplementation [33]. It may thus be prudent to suggest regular screening for kidney function, nephrocalcinosis and nephrolithiasis in affected patients.

Azole agents (e.g., fluconazole or ketoconazole) and rifampin have not been used to treat hypercalcemia during pregnancy and postpartum in patients with pathogenic CYP24A1 mutations, but have been suggested as potential candidates for long-term treatment in this setting [9]. Azole agents that are usually used to treat fungal infections are inhibitors of cytochrome P450-enzymes that also inhibit 1-alpha-hydroxylase (cytochrome-P450 family 27 subfamily B member 1; CYP27B1) and thus the conversion (activation) of 25(OH)D to 1,25(OH)_2_D. Case reports describe their successful use in patients with pathogenic CYP24A1 mutations [9,34]. Fluconazole may be preferred over ketoconazole due to generally fewer side effects, in particular less hepatotoxicity. The tuberculosis drug rifampin (also termed rifampicin) has also been successfully used to treat patients with pathogenic CYP24A1 mutations [35]. This drug induces the enzyme cytochrome-P450 family 3 subfamily A member 4 (CYP3A4) that inactivates vitamin D metabolites by an alternative degradative pathway to CYP24A1. Consequently, it has been recommended to avoid use of medications and foods (e.g., starfruit, pomegranate, and white grapefruit) that can inhibit CYP3A4 in patients with pathogenic CYP24A1 mutations. Rifampin has an excellent safety profile but may also be hepatotoxic in some cases [35]. Fluconazole and rifampin might be considered for treatment of hypercalcemia during pregnancy as there is much experience with these drugs in treating infections in pregnant women, but this requires careful consideration of the risks and benefits for the individual patient [36,37].

Given that the prevalence of pathogenic CYP24A1 mutations is very low and considering the beneficial effects of preventing and treating vitamin D deficiency, we do support the current recommendations for vitamin D intakes and supplementation [31,38,39,40,41,42]. The existing risk of severe hypercalcemia with high vitamin D bolus doses as it has been applied for rickets prevention in the UK in the 1950s or in Poland and former East Germany in the 1980s, in individuals with pathogenic CYP24A1 mutations, points towards caution with high dose vitamin D supplementation [33]. Awareness must be increased among clinicians that in the event of hypercalcemia with low PTH concentrations and in patients with nephrolithiasis and/or nephrocalcinosis, pathogenic CYP24A1 mutations should be considered as a potential underlying disease. If pathogenic CYP24A1 mutations are suspected, the measurement of vitamin D metabolites, i.e., the 24,25(OH)_2_D_3_ to 25(OH)D_3_ ratio (or vice versa) is recommended and if it is pathologic in terms of relatively low 24,25(OH)2D_3_ concentrations for the prevailing 25(OH)D_3_ status, genetic analyses should be performed to establish the diagnosis. In general, individuals without pathogenic CYP24A1 mutations do have a 25(OH)D_3_ to 24,25(OH)_2_D_3_ ratio >25 and those with pathogenic mutations do have a respective ratio of > 80 [43,44]. Of note, there are different approaches regarding calculations and cut-offs for this ratio published, pointing to the need for harmonization of this issue in the future [45,46,47].

It is a limitation of our work that it was not based on an a priori registered systematic review, but we consider it as a main strength that our paper is the first to specifically address the issue of pathogenic CYP24A1 mutations in pregnancy. We are aware that our conclusions are based on observational data with all their inherent limitations, but we do hope that our work may provide some guidance for clinicians regarding the management of pregnant women with pathogenic CYP24A1 mutations.

## 5. Conclusions

Pregnant women with pathogenic CYP24A1 mutations are at particular high risk of hypercalcemia and therewith associated complications. Minimizing vitamin D supply in these women seems to be highly effective as a therapeutic approach, whereas the existing data are insufficient and limited to draw firm conclusions on specific drug treatments in affected women. It seems reasonable to screen for serum calcium concentrations in pregnant women in order to accurately diagnose and treat cases with pathogenic CYP24A1 mutations and other disorders of calcium metabolism. Affected women can be informed that the most likely outcome of their pregnancy is that they will have a healthy infant, but ionized calcium should be measured in the newborns. Long-term screening with reference to kidney function and nephrolithiasis seems to be reasonable in patients with pathogenic CYP24A1 mutations, but whether any chronic drug treatment should be advised requires further investigations.

## Figures and Tables

**Table 1 nutrients-14-02518-t001:** Selected characteristics of the case patient with pathogenic CYP24A1 mutations at the endocrine outpatient clinic.

Parameter (Unit)	Reference Range	1st Visit	2nd Visit	3rd Visit	4th Visit	5th Visit
Gestational week		24	27	31	33	Two months after giving birth
Albumin adjusted serum calcium (mmol/L)	2.20 to 2.65	3.08	3.00	3.07	2.95	2.35
Ionized serum calcium (mmol/L)	1.15 to 1.35	1.57	1.51	1.57	1.50	1.29
Total serum calcium (mmol/L)	2.20 to 2.65	2.97	2.93	2.99	2.84	2.58
Serum phosphate (mmol/L)	0.84 to 1.45	0.74	0.72	0.89	0.95	1.19
Serum magnesium (mmol/L)	0.70 to 1.10	0.55	0.55	0.55	0.66	0.74
Serum creatinine (mg/dL)	up to 1.00	0.78	0.82	0.88	0.75	1.02
eGFR (CKD-EPI) (ml/min/1.73 m^2^)	90 to 120	106	100	92	111	76
Spot urine calcium/creatinine ratio (mmol/mmol)	up to 0.60	1.02	0.82	0.25	0.54	0.29
Parathyroid hormone (pg/mL)	15.0 to 65.0	8.0	7.1	7.6		8.4
25-hydroxyvitamin D (nmol/L) *	75 to 150	87	75	75		45
1.25-dihydroxyvitamin D (pmol/L) *	52 to 267	279	325	295		73
Bone-specific alkaline phosphatase (µg/L)	4.7 to 27.0	6.9				19.0
Osteocalcin (ng/mL)	1.0 to 35.0	19.3	24.3	29.6		42.6
Procollagen type 1 N-terminal propetide (ng/mL)	15 to 49	50.0	81.3	87.9		94.4
C-terminal telopeptide of type 1 collagen (ng/mL)	0.03 to 0.37	0.29	0.35	0.67		0.55
Fibroblast-growth-factor-23 (pg/mL)	14.0 to 48.0		176.1	156.6		56.0
Parathyroid hormone-related peptide (pmol/L)	0.0 to 1.3	1.2	1.2	0.5		
25-hydroxyvitamin D_3_ (nmol/L) **	NA			90.2		71.7
25-hydroxyvitamin D_2_ (nmol/L) **	NA			1.8		2.7
25-hydroxyvitamin D_2_ + D_3_ (nmol/L) **	75 to 150			92.0		74.4
24,25-dihydroxyvitamin D_3_ (nmol/L) **	NA			0.66		0.13
24,25-hydroxyvitamin D_3_ to 25-hydoxyvitamin D_3_ ratio (%) **	>3			0.73		0.18

eGFR (CKD-EPI: estimated glomerular filtration rate (Chronic Kidney Disease Epidemiology Collaboration); * measured by immunoassays; ** measured by a liquid chromatography tandem mass spectrometry (LC-MS/MS) method.

**Table 2 nutrients-14-02518-t002:** Characteristics of pregnancies of women with pathogenic CYP24A1 mutations.

Case Number of the Mother	Reference Number	Age (Years)	Number of Fetuses	Peak Serum Calcium in Pregnancy (mmol/L) *	Major Maternal Pregnancy Complications	Type of Delivery **	Major Maternal Postpartum Complications	Live Birth	Breast-Feeding	Major Newborn Complications
1	[15]	27	1	Not reported	Pre-eclampsia, polyhydramnios	Caesarian section	Acute kidney injury	Yes	Yes	Symptomatic hypercalcemia at 5 days
	[16]	32	1	Ionized serum calcium > 1.5	Hypertension, worsening renal function	Caesarian section	Worsening renal function	Yes	No	Symptomatic hypocalcemia at 3 months
2	[17]	23	1	NA	None	NA	Pre-eclampsia, hypercalcemic crisis	Yes	NA	Convulsions, hypoglycemia, necrotizing enterocolitis
		NA	1	2.89	Hypertension	Vaginal	Not reported	Yes	NA	Hypoglycemia, hypercalcemia
		NA	1	3.44	Symptomatic hypercalcemia	Vaginal	None	Yes	NA	Hypercalcemia
		NA	1	2.88	None	Vaginal	Hypertension, hypercalcemic crisis	Yes	NA	Hypoglycemia
		NA	1	2.92	Hypertension	Vaginal	Hypercalcemic crisis	Yes	NA	None
3	[17]	21	1	2.87	None	Vaginal	Hypercalcemia	Yes	NA	Hypercalcemia
		NA	1	2.83	Hypertension	NA	Hypercalcemia	Yes	NA	Hypercalcemia
4	[18]	47	2	3.11	Hypertension, diabetes	Caesarian section	Hypercalcemia	Yes for both	No	None
5	[18]	36	2	NA	NA	NA	Hypercalcemic crisis	Yes for both	NA	None
		NA	1	NA	None	NA	None	Yes	NA	None
		NA	1	NA	None	NA	None	Yes	NA	None
6	[19]	32	1	3.27	Pre-eclampsia	Caesarian section	Hypertension, acute kidney injury	Yes	NA	Mild hypercalcemia
7	[19]	32	NA	NA	Nephrolithiasis	NA	NA	NA	NA	NA
8	[20]	20	2	3.07	Hypertension	Vaginal	Hypercalcemic crisis, acute pancreatitis	No	No	Not alive (intrauterine demise at 26 weeks)
		20	1	2.87	Acute pancreatitis	Vaginal	None	Yes	No	None
9	[21]	33	2	3.4	Hypertension	Caesarian section	NA	Yes for one	NA	Development disorder, anorectal malformation, asymptomatic hypercalcemia
10	[25]	20	2	3.82	Acute pancreatitis	NA	Acute pancreatitis	No	No	Not alive (intrauterine demise at 26 weeks)
		20	1	2.99	Pre-eclampsia, acute pancreatitis	Vaginal	None	Yes	No	None
11	[22]	24	NA	3.07	NA	NA	NA	NA	NA	NA
		26	NA	3.07	NA	NA	NA	NA	NA	NA
		27	NA	2.92	NA	NA	NA	NA	NA	NA
		28	1	3.04	Pre-eclampsia	NA	NA	Yes	Yes	Slight hypocalcemia
12	[23]	NA	1	2.92	Intrauterine growth retardation	NA	NA	Yes	NA	None
		NA	1			NA	NA	Yes	NA	None
		35	2	2.99	Rupture of membranes	Vaginal and caesarian section	Hypercalcemic crisis	Yes	NA	None
13	[24]	Mid 20	1	3.3	Idiopathic cholestasis	NA	NA	Yes	NA	None
		End 20	1	2.6	Idiopathic cholestasis	NA	NA	Yes	NA	None

NA, not available; * Total or albumin adjusted serum/plasma calcium is indicated; ** Most cases with NA may have had vaginal delivery but this was often not clearly indicated; Hypercalcemic crisis was usually defined as serum calcium of at least 3.5 mmol/L plus symptoms.

**Table 3 nutrients-14-02518-t003:** Specific drug treatments of women with pathogenic CYP24A1 mutations during pregnancy.

Case Number of the Mother	Reference Number	Intravenous Hydration	Loop Diuretics	Glucocorticoids	Phosphate Supplements	Calcitonin
1	[15]	No *	No	No	No	No
	[16]	**Yes**	No	**Yes**	**Yes**	**Yes**
2	[17]	No	No	No	No	No
		No	No	No	No	No
		No	**Yes**	No	No	No
		No	No	No	No	No
		No	No	No	No	No
3	[17]	No	No	No	No	No
		No	No	No	No	No
4	[18]	**Yes**	**Yes**	**Yes**	No	No
5	[18]	No	No	No	No	No
		No	No	No	No	No
		No	No	No	No	No
6	[19]	**Yes**	No	**Yes**	No	No
7	[19]	No	No	No	No	No
8	[20]	No	No	No	No	No
		**Yes**	No	No	**Yes ****	**Yes**
9	[21]	No	**Yes**	No	No	No
10	[25]	No	No	No	No	No
		No	No	No	No	No
11	[22]	No	No	No	No	No
		No	No	No	No	No
		No	No	No	No	No
		No	No	No	No	No
12	[23]	No	No	No	No	No
		No	No	No	No	No
		No	No	No	No	No
13	[24]	No	No	**Yes**	No	No
		No	No	No	No	No

* If not specifically indicated in the manuscript we assumed no treatment if it was not described; ** Omeprazole was also prescribed to reduce intestinal calcium absorption.

**Table 4 nutrients-14-02518-t004:** Specific drug treatments of women with pathogenic CYP24A1 mutations postpartum.

Case Number of the Mother	Reference Number	Intravenous Hydration	Loop Diuretics	Glucocorticoids	Potassium Supplements	Calcitonin	Denosumab	Bisphosphonates
1	[15]	No *	No	No	No	No	No	No
	[16]	No	No	No	No	**Yes**	No	No
2	[17]	**Yes**	**Yes**	No	No	**Yes**	No	**Yes**
		No	No	No	No	No	No	No
		No	**Yes**	No	No	No	No	**Yes**
		**Yes**	No	No	No	No	No	No
		**Yes**	No	No	No	No	No	No
3	[17]	**Yes**	No	No	No	No	No	No
		**Yes**	No	No	No	No	No	No
4	[18]	**Yes**	**Yes**	**Yes**	No	No	**Yes**	No
5	[18]	**Yes**	No	**Yes**	No	No	**Yes**	**Yes**
		No	No	No	No	No	No	No
		No	No	No	No	No	No	No
6	[19]	**Yes**	No	**Yes**	No	No	No	No
7	[19]	No	No	No	No	No	No	No
8	[20]	**Yes**	No	**Yes**	No	**Yes**	No	No
		No	No	No	No	**Yes**	No	No
9	[21]	No	No	No	No	No	No	No
10	[25]	**Yes**	No	No	No	**Yes**	No	No
		No	No	No	No	No	No	No
11	[22]	No	No	No	No	No	No	No
		No	No	No	No	No	No	No
		No	No	No	No	No	No	No
		No	No	No	No	No	No	No
12	[23]	No	No	No	No	No	No	No
		No	No	No	No	No	No	No
		**Yes**	No	**Yes**	No	No	No	**Yes**
13	[24]	No	No	No	No	No	No	No
		No	No	No	No	No	No	No

* If not specifically indicated in the manuscript we assumed no treatment if it was not described.

## Data Availability

All available data are presented in the manuscript.

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
