# Peer review of "Hypercalcemia in Pregnancy Due to CYP24A1 Mutations: Case Report and Review of the Literature"

_nutrients, 2022, doi:10.3390/nu14122518_

Round 1

Reviewer 1 Report

The present review offers a very practical guidance on the management of hypercalcemia in pregnancy secondary to CYP24A1 mutation, with the very strategy being minimal to no vitamin D supplementation. The review reads well and the case is well presented. The authors should explicitly say that the review provided is narrative (if this is the case) as its limitations inherent in not doing a priori registered review were already duly acknowledged in the limitations. The paper will nevertheless likely benefit if a PRISMA flowchart is provided.

Author Response

We thank the reviewers for their valuable comments on our manuscript. We revised our work accordingly and present a point by point reply to the reviewer comments below.

Reviewer 1

The present review offers a very practical guidance on the management of hypercalcemia in pregnancy secondary to CYP24A1 mutation, with the very strategy being minimal to no vitamin D supplementation. The review reads well and the case is well presented. The authors should explicitly say that the review provided is narrative (if this is the case) as its limitations inherent in not doing a priori registered review were already duly acknowledged in the limitations. The paper will nevertheless likely benefit if a PRISMA flowchart is provided.

Response: We thank the reviewer for this positive comment. As suggested by the reviewer we now mention in the Abstract that this is a narrative review as it does not fulfill all criteria for systematic reviews including an a priori registration of the literature search and review. Although we agree with the reviewer that a PRISMA flowchart is an integral part of a systematic review, we refrained from including a PRISMA flowchart in order to avoid confusions regarding the fact that this work was not an a priori registered systematic review. If the reviewer, however, insists to include a PRISMA flowchart we would, of course, be willing to do this.

Reviewer 2 Report

Previously, mutations in CYP24A1 have been identified as causative in the etiology of idiopathic infantile hypercalcemia. This gene encodes a mitochondrial enzyme responsible for metabolism of Vitamin D products. In this case study and literature review the authors discuss that individuals with this mutation are not just at risk of hypercalcemia in infancy but throughout the lifespan and that it may manifest during pregnancy. They present the case of a pregnant patient with this mutation and associated hypercalcemia and treatment and utilize information from the literature to provide information on management of CYP24A1 mutations and the associated hypercalcemia in pregnancy.

A few minor editing suggestions are noted below: 

I would suggest removing all dates from the manuscript, especially the birth date in the case study. It is not necessary for the reader and could be considered identifying information. Also remove the dates from Table 1. 

Remove the word “systematic” from line 146 as this is not a systematic review.

In Table 3, the columns for denosumab and bisphosphonates can be removed since no patients were treated with these drugs and especially since there is a statement to that effect already included in the manuscript.

In the Discussion (starting at line 122) the authors state that it is effective to advise discontinuation of Vitamin D supplements. However, as seen in Table 1, none of the parameters studied were improved in their patient who was given this advice. Perhaps a line to that effect could be included in the discussion. 

Author Response

We thank the reviewers for their valuable comments on our manuscript. We revised our work accordingly and present a point by point reply to the reviewer comments below.

Reviewer 2

Previously, mutations in CYP24A1 have been identified as causative in the etiology of idiopathic infantile hypercalcemia. This gene encodes a mitochondrial enzyme responsible for metabolism of Vitamin D products. In this case study and literature review the authors discuss that individuals with this mutation are not just at risk of hypercalcemia in infancy but throughout the lifespan and that it may manifest during pregnancy. They present the case of a pregnant patient with this mutation and associated hypercalcemia and treatment and utilize information from the literature to provide information on management of CYP24A1 mutations and the associated hypercalcemia in pregnancy.

Response: We thank the reviewer for the review of our manuscript and, of course, agree with the above mentioned summary and conclusions.

A few minor editing suggestions are noted below: 

I would suggest removing all dates from the manuscript, especially the birth date in the case study. It is not necessary for the reader and could be considered identifying information. Also remove the dates from Table 1. 

Response: We agree with the reviewer that the dates could be considered as identifying information and we therefore removed all precise dates just mentioning the seasons and year.

Remove the word “systematic” from line 146 as this is not a systematic review.

Response: We agree with this comment and have removed the word systematic from line 146.

In Table 3, the columns for denosumab and bisphosphonates can be removed since no patients were treated with these drugs and especially since there is a statement to that effect already included in the manuscript.

Response: We agree with this comment and removed the respective columns in Table 3.

In the Discussion (starting at line 122) the authors state that it is effective to advise discontinuation of Vitamin D supplements. However, as seen in Table 1, none of the parameters studied were improved in their patient who was given this advice. Perhaps a line to that effect could be included in the discussion.

Response: We thank the reviewer for carefully reading our manuscript. We agree that despite our treatment and advice there were no major changes in serum calcium concentrations, but we noticed a significant decrease in urinary calcium excretion in the patient. Data interpretation is, however, challenging as we, of course, have no data on the natural course of the disease without any intervention. This is now mentioned in the manuscript as follows: “Notably, despite our advice to minimize vitamin D supply in our case patient there were no major changes in serum calcium concentrations, but we noticed a significant decrease in urinary calcium excretion. Data interpretation of this single case is, however, challenging as we, of course, have no data on the natural course of the disease without any intervention”. We hope that the reviewer agrees with this.

Reviewer 3 Report

Hypercalcemia in pregnancy can have consequences for the pregnant patient and the fetus. The association with the CYP24A1 genetic mutation is an exciting topic to study. The case described is among the few that address this pathology in pregnancy. The manuscript is clear, relevant to the field, and presented in a well-structured manner.

The authors made a coherent and detailed description of the case in the first part. Therefore, in this section, one table is appropriate and easy to interpret and understand.

The review is clear, comprehensive, and relevant to the field. I did not identify any knowledge gap. I identified a review[1] presenting data related to hypercalcemia due to CYP24A1 mutations in the general population, including pregnant women. This article is included in the list of references.

This current review is still relevant and of interest to the scientific community because it addresses a particular population. The cited references are older than five years but still relevant. In addition, there are more citations, including the first author Pilz S.

The statements and conclusions are drawn coherently and supported by the listed citations.

The tables are appropriate, properly show the data, and they are easy to interpret and understand. I appreciate that the data are interpreted appropriately and consistently throughout the manuscript.

The Discussions section draws a parallel between the data obtained in the situation of the presented case and the results of other publications.

The conclusions are consistent with the evidence and arguments presented. In addition, the article states that the Ethics Committee approved the study at the Medical University of Graz, Austria. Unfortunately, no details refer to the data availability statement or the approval number.

I have some observations related to the abbreviations. There are some which are not explained in the article: CYP27B1, CYP24A1, RCT. 

I recommend establishing a form and writing, in the same way, the abbreviations—for example, 25-hydroxyvitamin D3 and 25-hydroxyvitamin D3.

1.Cappellani, D., Brancatella, A., Morganti, R., Borsari, S., Baldinotti, F., Caligo, M. A., Kaufmann, M., Jones, G., Marcocci, C., & Cetani, F. (2021). Hypercalcemia due to CYP24A1 mutations: a systematic descriptive review. European journal of endocrinology186(2), 137–149. https://doi.org/10.1530/EJE-21-0713

Author Response

We thank the reviewers for their valuable comments on our manuscript. We revised our work accordingly and present a point by point reply to the reviewer comments below. 

Reviewer 3

Hypercalcemia in pregnancy can have consequences for the pregnant patient and the fetus. The association with the CYP24A1 genetic mutation is an exciting topic to study. The case described is among the few that address this pathology in pregnancy. The manuscript is clear, relevant to the field, and presented in a well-structured manner.

The authors made a coherent and detailed description of the case in the first part. Therefore, in this section, one table is appropriate and easy to interpret and understand.

The review is clear, comprehensive, and relevant to the field. I did not identify any knowledge gap. I identified a review[1] presenting data related to hypercalcemia due to CYP24A1 mutations in the general population, including pregnant women. This article is included in the list of references.

This current review is still relevant and of interest to the scientific community because it addresses a particular population. The cited references are older than five years but still relevant. In addition, there are more citations, including the first author Pilz S.

The statements and conclusions are drawn coherently and supported by the listed citations.

The tables are appropriate, properly show the data, and they are easy to interpret and understand. I appreciate that the data are interpreted appropriately and consistently throughout the manuscript.

The Discussions section draws a parallel between the data obtained in the situation of the presented case and the results of other publications.

The conclusions are consistent with the evidence and arguments presented. In addition, the article states that the Ethics Committee approved the study at the Medical University of Graz, Austria. Unfortunately, no details refer to the data availability statement or the approval number.

I have some observations related to the abbreviations.

Response: We thank the reviewer for this positive comment.

I have some observations related to the abbreviations.

There are some which are not explained in the article: CYP27B1, CYP24A1, RCT.

Response: We have adopted this as suggested by the reviewer. In detail, we now mention that CYP24A1 is the cytochrome-P450 family 24 subfamily A member 1 and that CYP27B1 is the cytochrome-P450 family 27 subfamily B member 1, and that RCT is the abbreviation for randomized controlled trial.

I recommend establishing a form and writing, in the same way, the abbreviations—for example, 25-hydroxyvitamin D3 and 25-hydroxyvitamin D3.

1.Cappellani, D., Brancatella, A., Morganti, R., Borsari, S., Baldinotti, F., Caligo, M. A., Kaufmann, M., Jones, G., Marcocci, C., & Cetani, F. (2021). Hypercalcemia due to CYP24A1 mutations: a systematic descriptive review. European journal of endocrinology186(2), 137–149. https://doi.org/10.1530/EJE-21-0713

Response: According to the suggestion of the reviewer we have now changed the abbreviations as in the mentioned reference.